


**Investigation of near-global daytime boundary layer height**
**using high-resolution radiosondes: First results and**
**comparison with ERA-5, MERRA-2, JRA-55, and NCEP-2**
**reanalyses**
Jianping Guo[a], Jian Zhang[b], Kun Yang[c], Hong Liao[d], Shaodong Zhang[e], Kaiming
Huang[e],Yanmin Lv[a], Jia Shao[f], Tao Yu[b], Bing Tong[a], Jian Li[a], Tianning Su[g], Steve
H.L. Yim[h,i], Ad Stoffelen[j], Panmao Zhai[a], and Xiaofeng Xu[k]
[a] State Key Laboratory of Severe Weather, Chinese Academy of Meteorological
Sciences, Beijing 100081, China
[b] Hubei Subsurface Multi-scale Imaging Key Laboratory, Institute of Geophysics and
Geomatics, China University of Geosciences, Wuhan 430074, China
[c] Department of Earth System Science, Tsinghua University, Beijing 100084, China
[d] Nanjing University of Information Science and Technology, Nanjing 210044, China
[e] School of Electronic Information, Wuhan University, Wuhan 430072, China
[f] College of Informatics, Huazhong Agricultural University, Wuhan 430070, China
[g] Department of Atmospheric and Oceanic Sciences, University of Maryland, College
Park, Maryland 20740, USA
[h] Department of Geography and Resource Management, The Chinese University of
Hong Kong, Shatin, Hong Kong, China
[i] Stanley Ho Big Data Decision Analytics Research Centre, The Chinese University of
Hong Kong, Shatin, Hong Kong, China
[j] The Royal Netherlands Meteorological Institute (KNMI), 3730 AE De Bilt, The
Netherlands
[k] China Meteorological Administration, Beijing 100081, China
Correspondence to:
Dr. Jian Zhang (Email: zhangjian@cug.edu.cn)



# Abstract

The planetary boundary layer height (BLH) governs the vertical transport of mass,
momentum and moisture between the surface and the free atmosphere, and thus its
characterization is recognized as crucial for air quality, weather and climate. Although
reanalysis products can provide important insight into the global view of BLH in a
seamless way, the in situ observed BLH on a global scale remains poorly understood
due to the lack of high-resolution (1-s or 2-s) radiosonde measurements. The present
study attempts to establish a near-global BLH climatology at synoptic times (0000 and
1200 UTC) and in the daytime using high-resolution radiosonde measurements over
300 radiosonde sites worldwide for the period 2012 to 2019, which is then compared
against the BLHs obtained from four reanalysis datasets, including ERA-5, MERRA-2,
JRA-55, and NCEP-2. The variations of BLH exhibit large spatial and temporal
dependence, and as a result the BLH maxima are generally discerned over the regions
such as Western United States and Western China, in which the balloon launch times
mostly correspond to the afternoon. The diurnal variations of BLH are revealed with a
peak at 1700 Local Solar Time (LST). The most promising reanalysis product is ERA-
5, which underestimates BLH by around 130 m as compared to radiosondes. In addition,
MERRA-2 is a well-established product and has an underestimation of around 160 m.
JRA-55 and NCEP-2 might produce considerable additional uncertainties, with a much
larger underestimation of up to 400 m. The largest bias in the reanalysis data appears
over the Western United States and Western China and it might be attributed to the
maximal BLH in the afternoon when the boundary layer has grown up. Statistical
analyses further indicate that the biases of reanalysis BLH products are positively
associated with orographic complexity, as well as the occurrence of static instability.
To our best knowledge, this study presents the first near-global view of high-resolution
radiosonde derived BLH and provides a quantitative assessment of the four frequently
used reanalysis products.
**Keywords**. Radiosonde; boundary layer height; reanalysis; sensible heat flux



## 1. Introduction

The planetary boundary layer (PBL) and its evolution has a profound influence on research fields such as air quality (Stull, 1988; Li *et al.,* 2017), boundary layer cloud and fog (Liu and Liang, 2010), convective storm (Oliveira *et al*., 2020) and global warming (Davy and Esau, 2016), among others. It is well known to be influenced by radiative cooling at night and by downward solar radiation reaching the ground surface at daytime, respectively, forming a stable boundary layer (SBL) and convective boundary layer (CBL), with a typical boundary layer depth (BLH) of less than 500 m and 1–3 km (Zhang *et al.,* 2020a), respectively. For climate models, most of the PBL processes occur at sub-grid scales and thus are either underrepresented or not fully represented (von Engeln and Teixeira, 2013). Meanwhile, there are many problems in elucidating the PBL processes using numerical model simulations (Martins *et al.,* 2010), even over the relatively homogeneous ocean (Belmonte and Stoffelen, 2019), which is likely due to the scarcity of fine-scale vertical observations of the atmosphere.

Over the oceans Belmonte and Stoffelen (2019) performed a climatological comparison between state-of-the-art reanalysis and scatterometer surface winds in the PBL, revealing mean and transient PBL model errors. Houchi *et al.* (2010), based on high-resolution radiosondes, verified the climatological wind profiles and found in particular a factor of 2–3 lower wind shear simulated by the European Centre for Medium-Range Weather Forecasts (ECMWF) model. Wind shear is recognized to be able to significantly modulate turbulent mixing of atmospheric pollutants (Zhang *et al.,* 2020b), and thus the inabilities of the model in this regard may have repercussions for air quality prediction.

The temporal and spatial variations in BLH have been extensively assessed in previous studies at a regional or national scale, such as the contiguous United States (Seidel *et al.,* 2012; Zhang *et al.,* 2020a), Europe (Palarz *et al*., 2018), China (Guo *et al.,* 2016; Zhang *et al*., 2018, Su *et al*., 2018), Arctic and Antarctic (Zhang *et al*., 2011), which are mainly implemented by radiosonde measurements, reanalysis or both. And



notable diurnal and seasonal cycles have been revealed (e.g., Guo *et al.,* 2016; Short *et*
*al.,* 2019). Besides the regional results, several attempts have been made to provide
global-scale retrievals of BLH using the Global Positioning System radio occultation
(GPS RO) and Integrated Global Radiosonde Archive (IGRA) version 2 (Seidel *et al.,*
2010; Gu *et al.,* 2020; Ratnam and Basha, 2010), in which seasonal variations and
maritime-continental contrasts of BLHs have been achieved. The measurements of GPS
RO, at a vertical resolution of 100 m around the PBL top, are typically used to determine
BLH by searching for the altitude with a sharp gradient in the refractivity profile (Basha
*et al.,* 2018). However, such sharp gradient of refractivity might overestimate BLH
compared to other methods that the community usually used, such as the parcel method
(Seidel *et al.,* 2010). Compared with high-resolution soundings, IGRA is sparsely
sampled in the vertical, which could result in large uncertainties in estimating BLH.
Likewise, additional errors could be introduced in reanalysis products for their sparse
vertical resolutions, which are equivalent to or bigger than IGRA. A large spread
emerges in the explicit determination of BLH from a variety of instruments, in spite of
that the BLH detection based on radiosonde is the most accepted methodology for
deriving CBL and SBL (Seidel *et al.,* 2012; de Arruda Moreira *et al.,* 2018).

A wide range of reanalysis products, such as those from the fifth generation

ECMWF atmospheric reanalysis of the global climate (ERA-5), the National
Aeronautics and Space Administration (NASA) Modern-Era Retrospective-analysis for
Research and Applications version 2 (MERRA-2), Japanese 55-year Reanalysis (JRA-
55), and the NCEP climate forecast system version 2 (NCEP-2), provide a rich
ensemble of climate data products (Saha *et al.,* 2014; Hersbach *et al.,* 2020; Kobayashi
*et al.,* 2015; Gelaro *et al.,* 2017), but are sensitive to both empirical parameterizations
and the diagnostic method chosen, while verification by direct observations of BLH are
sparse (Seibert *et al.,* 2000). Some inter-comparisons between instruments, such as
radiosonde, LIDAR, and ERA-interim reanalysis have been previously conducted, and
a rough consistency has been yielded (e.g., Guo *et al.,* 2016; Korhonen *et al.,* 2017;
Zhang *et al.,* 2016). However, Basha *et al.* (2018) demonstrate that ERA-interim can



underestimate BLH by around 900 m compared to GPS RO. This underestimation may
be caused by different kinetic or thermodynamic assumptions use. For instance, ERA-
interim is implemented with a bulk Richardson number method (Palm *et al.*, 2005),
which is believed to be suitable for all atmospheric conditions (Anderson, 2009). It is
worth highlighting that the state-of-art reanalysis could be one of the most promising
data sources for obtaining the synoptic or climatological features of BLH.

Despite much progress made in developing the BLH products, there are still some

unresolved issues in quantifying the variability of BLH from a global perspective.
These issues include: the worldwide variation of BLH by high-resolution vertical
soundings, the inter-comparisons among reanalysis datasets, and further evaluations
with radiosonde observations, especially in the daytime based on the same retrieval
algorithm. To this end, this study seeks to address the following scientific questions: (1)
a climatological distribution of near-global BLH by using high-resolution radiosonde
measurements; (2) inter-comparisons of ERA-5, MERRA-2, JRA-55, and NCEP-2 with
additional evaluation with radiosondes; and (3) investigate potential sources for the
biases of BLH between observation and reanalysis. The rest of the paper is organized
as follows. The descriptions of high-resolution radiosonde data, reanalysis products,
and the bulk Richardson number method are given in Section 2. Section 3 presents the
spatial distributions of BLH by radiosonde and reanalyses and their inter-comparisons.
A brief conclusion and remarks are finally outlined in Section 4.
**2. Data descriptions and BLH retrieval method**
*2.1 High-resolution radiosonde measurements*

Until January 2018, IGRA provided atmospheric soundings at around 445

radiosonde sites across the globe, including pressure, temperature, humidity and wind.
The number of pressure levels below 500 hPa is around 10. By comparison, for high-
resolution radiosondes, the sampling rate is 1-s or 2-s, corresponding to a vertical
resolution of approximately 5–10 meters throughout the atmosphere. The high-



resolution radiosonde measurements used in the present study are obtained from 342
sites around the world, which are provided by several organizations, including the
China Meteorological Administration (CMA), the National Oceanic and Atmospheric
Administration (NOAA) of United States, the German Deutscher Wetterdienst (Climate
Data Center), the Centre for Environmental Data Analysis (CEDA) of United Kingdom,
the Global Climate Observing System (GCOS) Reference Upper Air Network
(GRUAN), and University of Wyoming.

The CMA maintains the China Radiosonde Network (CRN), which contains 120

operational stations homogeneously distributed across mainland China with a vertical
sampling rate of 1 second (5–8 m resolution), since 2011 (Guo *et al.,* 2016; 2019; Zhang
*et al.,* 2016; 2018; Su *et al.,* 2020). The NOAA started the Radiosonde Replacement
System (RRS) program in 2005, which involved 89 sites with a vertical resolution of 5
m (Zhang *et al.,* 2019). The German Deutscher Wetterdienst (Climate Data Center) has
been sharing the radiosonde measurements at 14 sites with a sampling rate of 2 seconds
since 2010. Moreover, the 10 m resolution soundings at 12 sites was provided by the
CEDA, which began to share soundings since 1990, and 8 radiosonde sites were shared
by GRUAN with a vertical resolution smaller than 10 m. An additional 93 sites came
from the University of Wyoming, which started in 2018, with a sampling rate of 2-s or
1-s. In total, over 678,000 soundings at 342 stations are used here for the period of
January 2012 to December 2019 in total of eight years, including 633,000 soundings at
the regular release times of 0000 and 1200 UTC and 43,000 more irregular observations
during intensive observation period (IOP).

Radiosonde measurements are taken twice per day following the World

Meteorological Organisation (WMO) protocol for synoptic times at 0000 and 1200
UTC (Seibert *et al.,* 2000), except for special field campaign observations at specified
stations or time ranges during IOPs. The protocol implies that stations at different
longitudes sample the diurnal cycle differently. For instance, stations near 0°E (London)
and 180°E (Samoa) sample at midnight and midday, while stations near 90°E
(Bangladesh) and 90°W (Chicago) sample at dawn and dusk, with intermediate





longitudes at linearly varying intermediate local solar times (LSTs) of day. For
wintertime regions near 90°W and 90°E, the release times are insufficient for evaluating
the BLH during daytime. Hence, the BLH estimates from regular radiosondes will vary
with longitude and season (McGrath-Spangler and Denning, 2012). Generally, the
principal PBL mechanism at night is associated with an SBL, which gradually
transitions into CBL in the morning (Stull, 1988; Zhang *et al.,* 2018). The transition
from SBL to CBL is generally quick and occurs swiftly after sunrise, but the reverse
process can be slow in the late evening (Taylor *et al.,* 2014). Despite the dominance of
CBL during the daytime, an SBL still occurs, especially in the event of overcast sky
(Zhang *et al.,* 2018; 2020) and near strong divergence in moist convective downbursts
(King *et al.,* 2017). To illustrate the daytime variation of BLH, we only selected the
soundings that are launched 2 hours after sunrise and 2 hours before sunset. The sunrise
and sunset times are gauged in a longitude bin size of 15 degrees and based on the
latitude of station and the calendar day of the release. As a result, 190,013 profiles
which include soundings launched at both synoptic times and during IOP, spanning
from January 2012 to December 2019, to obtain the BLH in the daytime. The spatial
distribution of file number for each site is displayed in Figure S1, in which the sites
with less than 10 matches are excluded.
*2.2 ERA-5, MERRA-2, JRA-55 and NCEP-2 reanalysis datasets*

ERA-5 is the successor of ERA-interim and undergo a variety of improvements,

including more recent parameterization schemes and data assimilation system, better
spatial resolution, both horizontally and vertically (137 levels), and improved
representation of evaporation balance, cyclones, soil moisture, and global precipitation
(Hersbach *et al.,* 2020). The BLH is composited in the ERA-5 product on a 1440×721
grids with 0.25° longitude and 0.25° latitude resolution. It is computed by the bulk
Richardson number method, with a temporal resolution of 1 hour.

MERRA-2 is the latest atmospheric reanalysis of the modern satellite era

produced by NASA's Global Modeling and Assimilation Office (GMAO). It includes
aerosol data assimilation, improvements on ozone, and cryospheric processes (Gelaro





*et al.,* 2017). The data is provided on a grid of 576×361 points with 0.625° longitude
and 0.5° latitude resolution and has 42 pressure levels (about 16 layers below 500 hPa),
with a temporal resolution of 3 h. In this product, the BLH is defined by identifying the
lowest level at which the heat diffusivity drops below a threshold value (McGrath-
Spangler and Denning, 2012). However, to preclude the uncertainty raised by different
methods adopted, the BLH by MERR-2 is extracted by bulk Richardson number
method, utilizing the parameters of horizontal wind, temperature, geopotential height,
relative humidity (RH), and surface pressure.
JRA-55 is the second Japanese global atmospheric reanalysis commissioned by
the Japan Meteorological Agency (JMA) (Kobayashi *et al.,* 2015). Data contains 37
pressure levels between 1 hPa and 1000 hPa (16 layers below 500 hPa), provided on a
grid of 288×145 points, with a horizontal spacing of 1.25°×1.25° and a temporal
resolution of 6 hours. The parameters, including geopotential height, temperature,
horizontal wind, surface pressure, and RH, are used to assess BLH as before. NCEP-2
has the coarsest model resolution than ERA-5 (Rinke *et al.,* 2019), with a spatial
resolution of 2.5° longitude and 2.5° latitude. The total level is 17 (6 layers below 500
hPa), which is substantially less than MERRA-2, JRA-55 or ERA-5, and the
temporal resolution is 6 hours. Similar parameters to JRA-55 are preserved to compute
BLH. It is noteworthy that all model times include 00 and 12 UTC and hence collocate
well with the synoptic radiosonde times.
*2.3 Normalized sensible heat flux in the daytime*
The sensible heat flux represents the level of energy that induces CBL growth (Wei
*et al.*, 2017), whereas the latent heat fluxes characterize the evaporation of moisture
from the soil to the CBL, which feedbacks on the development of CBL and the
formation of PBL cloud (Pal and Haeffelin, 2015) For a given amount of heat flux,
small latent heat fluxes usually mean more energy being available for PBL growth
(Chen *et al*., 2016). Moreover, the surface heat flux is closely associated with near-
surface meteorological variables. For instance, a lower RH usually indicates a larger
sensible heat flux and lower latent heat flux (Guo *et al*., 2019; Zhang *et al*., 2013).



Suppose that the heat supplied to the air at the radiosonde balloon launch time is the
area shaded under the heat flux curve (Fig.11.12 in Stull 1988), the normalized sensible
heat flux in the daytime is defined by
$$\overline{Q_H} \propto \int_{T_{sunrise}}^{T_{launch}} Q_H \rho^{-1} c_p^{-1} dt \qquad (1)$$

where $T_{sunrise}$ and $T_{launch}$ are the sunrise time and radiosonde balloon launch
time, $Q_H$ the sensible heat flux, $\rho$ the near-surface density and $c_p$ equals 1004
$J°C^{-1}kg^{-1}$. The similar principle is applied to the calculation of normalized latent
sensible heat flux as well.
*2.4 Bulk Richardson number method*

In the spirits of a like-for-like comparison, the BLHs derived from radiosonde and

reanalysis data (MERRA-2, JRA-55, and NCEP-2) are calculated using the bulk
Richardson number (BRN), which also serves as the built-in algorithm in ERA-5 for
BLH products. The BRN, an algorithm used to reflect how strongly buoyancy is
coupled to the vertical momentum (Scotti, 2015), has been widely used for the
climatological study of BLH from radiosonde measurements thanks to its applicability
and reliability for all PBL regimes (Anderson 2009; Seidel *et al.*, 2012; Guo *et al.*,
2019). It determines the BLH by identifying the level at which the bulk Richardson
number, represented by $Ri(z)$, reaches its critical value (Palm *et al.*, 2005) and is
formulated as

$$Ri(z) = \frac{\left(\frac{g}{\theta_{vs}}\right)(\theta_{vz}-\theta_{vs})z_{AG}}{(u_z-u_s)^2+(v_z-v_s)^2+(bu_*^2)} \qquad (2)$$

where g is the gravitational acceleration, $z_{AG}$ the height above ground level (AGL),
$\theta_v$ the virtual potential temperature, $u_*$ the surface friction velocity, and u and v
the horizontal wind components and $b$ a constant, which is usually set to zero due to
the fact that friction velocity is much weaker compared with the horizontal wind (Seidel
*et al.*, 2012). The subscripts of $z$ and $s$ denote the parameters at $z$ height above
ground and ground level, respectively.



256 It is known that $Ri(z)$ increases with increasing free flow stability (Zilitinkevich

257 and Baklanov, 2002). Below a critical value of 0.25, the flow is dynamically unstable

258 and likely cause turbulent motion. Nevertheless, since turbulence can also occur away

259 from this critical value (Haack *et al.,* 2014), care must be taken in that the critical value

260 might not be well defined, leading to uncertainty in estimating BLH. Meanwhile, the

261 BLH estimates were found not to change very much by differing the input of critical

262 values ($Ri = 0.2; 0.25; 0.3$) (Guo *et al.,* 2016). Therefore, for a given discrete $Ri$

263 profile, here we identify the BLH as the interpolated height at which the $Ri(z)$ firstly

264 crosses the critical value of 0.25 starting upward from the ground surface.

265 *2.5 Collocation procedure and a case study*

266 In contrast to the reanalysis data, the longitude, and latitude distributions of high-

267 resolution radiosonde are irregular. A precise comparison between reanalysis data and

268 sounding is required for consistency in time, latitude, and longitude. The matching

269 procedures implemented in this present study go as follows. (1) A latitudinal and

270 longitudinal matching procedure is carried out by finding the geographical grid cell of

271 the reanalysis product that contains the radiosonde station. (2) Time matching for ERA-

272 5 is to find the exact UTC time (hour) of the weather balloon launch. (3) For MERRA-

273 2, NCEP-2, and JRA-55 datasets, the requirement is to limit the time difference with

274 the weather balloon launch time to 1 hour.

275 A case at 0600 UTC 06 Jun 2016, Chongqing (29.6°N, 106.4°E) is shown in Figure

276 1. In this case, BLH obtained by sounding is 1,337 m and is closest to that by ERA-5,

277 which underestimates the height by 72 m. Compared with the radiosonde profile,

278 MERRA-2 can capture the main vertical structures and the magnitude of wind speed

279 (WS), RH, and temperature, but not the fine-scale vertical variations (Figure 1b). It also

280 slightly undervalues the BLH by 125 m. By and large, the profiles from JRA-55 are not

281 as accurate as those from MERRA-2. More specifically, the wind speed at some heights,

282 prominently above 2 km, is underestimated (Figure 1d); the mean RH is 4% less than

283 that from the sounding. As a result, JRA-55 substantially underestimates BLH by 399

284 m. The basic parameters outlined by NCEP-2, for instance, RH (5% larger than





sounding), temperature (3°C less than sounding), and wind speed (4.5 m/s larger than sounding), all have notable differences with the sounding (Figure 1c). The BLH is considerably underestimated by 729 m. Based on this case, we can note that the performances of ERA-5 and MERRA-2 are obviously better than those from JRA-55 and NCEP-2 in terms of the BLH, and that the remarkable underestimation by NCEP-2 can be attributed to the large error in the prediction of basic parameters, such as wind, temperature, and RH.

## 3. Results and discussion

*3.1 Overview of BLHs at two synoptic times*

The near-global mean BLHs at 0000 UTC from 2012 to 2019 by four reanalysis products are shown in Figure 2, in which the results obtained from radiosonde are overlaid by colored circles. The stations with sounding covering at least 2 continuous years are kept. The four reanalysis products yield an analogous result with respect to the spatial variation of BLHs, which are positively correlated with the sounding-derived BLH, with correlation coefficients of 0.90, 0.47, 0.46, 0.81 for ERA-5, NCEP-2, JRA-55, and MERRA-2, respectively. It is evident that the BLHs from NCEP-2 over the continents of Africa, Asia, and South America are 300 m thicker than those of the other three products (Figure 2b). Furthermore, the BLH in Antarctic by ERA-5 is notably 500 m lower than that by NCEP-2 and MERRA-2 (Figure 2a). Most of the mean BLHs by radiosonde are consistent with the reanalysis products, except that the values from all four reanalysis products over the Pacific Ocean and the contiguous U.S. are underestimated by about 300 m. Moreover, it is worth to note here that the BLHs by JRA-55 are considerably underestimated by around 1 km over these regimes. For 0000 UTC, the regions nearly from the east coast to the west coast of Pacific Ocean (UTC+8 to UTC+12, and UTC−12 to UTC−8) are covered by sunshine, and thus are filled with deeper PBL.



Comparable results at 1200 UTC are presented in Figure S2. Africa, the Middle
East, and the west of India and China, corresponding to local noon and afternoon, have
maximal BLHs of around 1.8 km. Moreover, it is noteworthy that the values from
NCEP-2 and JRA-55 over these areas are visibly lower than those from ERA-5 and
MERRA-2, particularly over Africa and the Middle East, whereas these low values can
barely be validated with soundings due to their sparse distribution. Over these areas,
the BLHs are underestimated by reanalysis by about 200 m relative to the sounding
results. Notably, BLHs from NCEP-2 over the continents of Africa are 1 km lower than
those from ERA-5 and MERRA-2. According to the results at 0000 and 1200 UTC, the
comparisons between reanalysis products and soundings demonstrate that the BLHs are
well resolved in the nighttime but are underestimated at daytime by reanalysis datasets.
For the near-global variation of BLH at a certain synoptic time, daytime and
nighttime appear on the map simultaneously, but as a function of longitude, which is
displayed in Figure 2. Thus, the variations at a fixed synoptic time on the map create a
picture of the diurnal BLH variation. Given the dominance of CBL in the daytime,
investigating the BLHs in the daytime is thus favorable for unravelling the underlying
causes for the discrepancies existed in the BLHs from both radiosonde and reanalysis.
Therefore, the following results show the variations of daytime BLH only, unless
otherwise noted.
*3.2 Variations over the day and comparisons with reanalysis products*
The climatological mean variations in the daytime BLH from the soundings and
four reanalysis products are drawn in Figure 3. The period spans from January 2012 to
December 2019 for most of the stations provided by China, the U.S., Germany, and the
U.K. As implied by the results from soundings (Figure 3e), the deepest PBL is observed
over the Tibetan Plateau (TP) and the northwest of China, the south of Africa, and the
west of U.S, with values as high as 1.7 km. The possible reason for this phenomenon is
that the weather balloons over these regions are basically launched in the early
afternoon of boreal summer (June–July–August) when the maximal BLH is usually
observed (Collaud Coen *et al.,* 2014; Guo *et al.,* 2016). The BLHs over the Pacific



Ocean are noticeably large, with values of 1.3 km. The longitudinal variation of BLH
is evident, likely due to LST variations of the soundings. Additionally, BLHs in the
middle and low latitudes are larger than high latitudes, which is consistent with the
findings in Gu *et al.* (2020).
By and large, the climatological results of BLH by radiosonde and four model
products are comparable, indicating that both capture the diurnal and seasonal
variations implied by the sounding LST times sampled. Among the model products,
ERA-5 shows the best prediction of BLH contrasted with radiosonde, with a correlation
coefficient of 0.88 (Figure 3a). Furthermore, the results from MERRA-2 are positively
correlated with those from the soundings, with a correlation coefficient of 0.66 (Figure
3b). The performances of JRA-55 and NCEP-2 are significantly poorer than those of
ERA-5 and MERRA-2, with correlation coefficients of 0.4 and 0.41, respectively
(Figure 3c, d). The values of BLH over the west of U.S and the west of China are
seriously underestimated by NCEP-2 and JRA-55 by around 800 m. Thus, we note that
ERA-5 and MERRA-2 are more robust in deriving the BLH, purely based on the
climatological distribution of BLHs.
Figure 4 illustrates the diurnal variations in BLH at 0000 and 1200 UTC and
during daytime. A notable diurnal variation can be noticed, with a minimum of 343 m
at 04 LST and a maximum of 1224 m at 17 LST (Figure 4a). The magnitude in BLH
during daytime are essentially larger than that at 0000 and 1200 UTC and has a maximal
value of 1926 m at 1700 LST (Figure 4b). It follows that some soundings that are
released at 0000 and 1200 UTC are excluded by the collocation procedure designed for
collecting samples in the daytime.
The radiosonde stations are mainly dispersed over the U.S, China, Austria, Europe,
the Pacifica Ocean, and the polar region, and only a few stations contribute over the
rest of the world. The polar region contains a station with a longitude larger/lower than
67.7°N/°W. Therefore, six regions are specifically examined in terms of the bias
between radiosonde and model product.



The BLH differences between radiosonde and ERA-5 are shown in Figure 5, in
which we specify the differences over the six above-mentioned regions. As observed in
Figure 5e, the BLH over most of the stations is underestimated to a slight extent, with
a near-global mean of 130.44 m. As expected, the most underestimated regions cover
the west of U.S, and southern China (Figure 5e), with a difference of around 200 m. In
addition, it is worth mentioning that the BLHs over the Pacific Ocean are overestimated
in four seasons, with a bias of around 400 m (Figure 5h). Among the six classified
regions, BLHs in Europe, East Asia, and polar are reliably determined by ERA-5, with
an average bias of around 50 m (Figure 5b, c, i). The bias seems to exhibit a seasonal
dependence, and it is larger in the warm seasons and smaller in the cool seasons.
Regardless of the small bias, the newest model product, ERA-5, properly estimates the
BLH, especially above the regions of Europe, the eastern U.S, East Asia, and polar.
Similarly, the BLHs by MERRA-2 are underestimated, with a near-global mean
bias of 159.72 m (Figure 6), which is slightly larger than that of ERA-5 (130.44 m).
This could indicate that the MERR-2-derived BLH is more dispersed than ERA-5. The
spatial distribution of bias value is broadly identical to that of ERA-5, except that the
BLHs over Europe, Austria, and polar region are well estimated by MERR-2, due to
much smaller mean biases at 42.10 m, 39.70/. m, and 52.27 m, respectively (Figure 6b,
g, i).
By comparison, the mean bias produced by JRA-55 is larger than those from ERA-
5 and MERRA-2, with a mean value of 352.59 m, as shown in Figure 7. The BLHs
above most stations are underestimated by JRA-55, particularly for the sites over
western China and western U.S, and the Pacific Ocean, with an underestimation of
about 800 m. The most underestimated stations cluster at the latitude range of 40–45°N,
with a mean difference of around 1 km (Figure 7f). Although the ensemble mean of
bias is significantly larger than ERA-5 and MERRA-2, the estimations over Europe and
the Polar regions seem to be acceptable, with mean values of 177.0 m and 99.2 m,
respectively (Figure 7b, i).





The mean bias by NCEP-2 is larger than that by JRA-55, with a mean value of
420.87 m, as illustrated by Figure 8. The distribution results are similar to JRA-55,
except for Europe and Austria, where the bias is about twice that of JRA-55.
In general, the comparison analysis of the daytime BLH results between soundings
and four reanalysis datasets indicates that ERA-5 reanalysis produces the BLH that is
closest to the high-resolution soundings. Interestingly, MERRA-2 can provide a good
spatial distribution of BLH. JRA-55 and NCEP-2 can only give a good prediction over
some regions, most of which tends to produce a much larger BLH estimates compared
to those from ERA-5 and MERRA-2.

*3.3 Correlations with near-surface meteorological variables and surface heat flux*

The PBL is the lowest part of the troposphere and evolves diurnally due to near-
surface thermodynamic variables through turbulent exchanges of momentum, heat, and
moisture (Pithan *et al*., 2015). Thus, the surface meteorological variables depend on the
underlying land surface and its coupling with the PBL, and they could act as a good
proxy for BLH under some specific circumstances (Zhang *et al*., 2013; Zhang *et al*.,
2018). An analysis of the correlation between the BLHs by radiosondes and near-
surface meteorological variables is presented in Figure 9. The variables include near-
surface air temperature at 2 m AGL ($T_{2m}$), pressure (Ps), RH, and WD, which are
extracted from the first level in sounding. The first level is assumed to be associated
with the near-surface variables (Serreze *et al*., 1992; Wang and Wang 2016). We note
that BLH, $T_{2m}$, RH and WD all have substantial diurnal and seasonal variability as
partly expressed in Eq. (2).
Relatively high positive (negative) correlation coefficients can be noticed between
BLH and $T_{2m}$ (RH), with mean values of 0.39/-0.51 (Figure 9a, c), implying that both
$T_{2m}$ and RH could be an adequate indicator for the temporal variation of BLH.
Moreover, the correlations between BLH and WD are also positively notable, with a
mean value of 0.24 (Figure 9d). By contrast, the correlation between Ps and BLH can
be ignored above most of the regions (Figure 9b).





The correlation analyses between BLH and normalized heat fluxes, which are
assessed by EAR-5 reanalysis products, are displayed in Figure 10. It is notable that
positive/negative correlation coefficients usually exist in normalized sensible/latent
heat flux, with a global mean of 0.29 and -0.31. This correlation is not high because
BLH also depends on the radiative heating/cooling and the temperature profile in
different stations (Yang *et al*., 2004).
For the climatological variation of BLH, the near surface variables such as $T_{2m}$,
RH and WS, and the normalized sensible/latent heat flux could be a good indicator.
Conversely, the development of BLH could also limit the magnitude of RH (McGrath-
Spanglerm, 2016).

*3.4 Potential sources for the bias between radiosonde and reanalysis products*

The possible sources for the difference between radiosonde and reanalysis could
be rather complicated. From the spatial pattern of BLH discrepancy results between
radiosonde and reanalysis (Figures 5–8), we can notice that the regions with large
differences tend to be observed over regions with high elevation, such as the TP in
China and Rocky mountain in the U.S. These regions generally have much more
complex orography. Coincidently, the soundings over the above-mentioned two regions
are all obtained from afternoon, in which the PBL develops to the maximum (Figure 4).
As expected, highest biases generally are accompanied with peak BLHs, which has also
been confirmed in our previous studies (cf. Figure 2c in Li *et al*., 2017). Therefore, the
biases depend on the LST when the weather balloon is launched, which at least could
not be ruled out.
In addition, the large differences primarily appear in the low and middle latitudes,
where thermal convection frequently occurs. Therefore, it is reasonable to infer that
static stability could exert an influence on the comparison results. Then, we will analyze
the probable influences from terrain and static stability on BLH differences.
We evaluate the influence from the orographic complexity around the sounding
station and calculate the standard derivation (STD) of elevation within 1°x1° grid, with
the help of 30 arc second digital elevation model (DEM) dataset. The analysis of the
correlation between the bias of the BLH and the standard derivation of the DEM is
shown in Figure 11. It follows that the influence from the orography appears
instrumental, given the correlation coefficient varying from 0.31 to 0.81. Furthermore,
the errors or uncertainties in ERA-5 are less easily impacted by the orographic
complexity due to the relatively lower correlation coefficient of 0.31 (Figure 11a).
Based on the correlation between orographic complexity (manifested by the STD
of the DEM) and the bias of a reanalysis relative to radiosonde measurements, it is
likely that the performances of MERR-2, JRA-55, and NCEP-2 might be restricted by
the complex underlying terrains. One of the reasons could be because global reanalysis
with coarse resolution that cannot resolve the sub-grid processes due to topography.
However, ERA-5 appears to be less dependent on terrain. In other words, the models
used in ERA-5 show sufficient capability and excellent performance in reproducing the
atmospheres, particularly in the PBL over complex terrains.
Lower tropospheric stability (LTS) is an indicator to describe the thermodynamic
state of the lower atmosphere and is defined by the differences in potential temperature
at 700 hPa and 1000 hPa (Guo *et al*., 2016). Typically, the smaller the LTS, the more
unstable the low troposphere. The mean LTS over each station is defined by the
ensemble mean by four reanalysis datasets, and its spatial distribution is depicted in
Figure 12. The lower troposphere over the western United States and western China is
more unstable compared to the rest of the world, with LTS of around 6K (Figure 11a),
which is likely associated with afternoon launch time of weather balloons. According
to the correlation between the bias of BLH and the mean LTS, it is clear that the
underestimation in BLH by JRA-55 and NCEP-2 products are negatively correlated
with LTS, with correlation coefficients of -0.32 and -0.36 (Figure 12b).
Besides the LTS, the role of lifted index could be another influential factor. The
lifted index is a predictor of latent instability (Galway, 1956), and it is defined as the
temperature difference between the environment temperature and an air parcel lifted
adiabatically at 500 hPa. The index is computed by the air temperature, RH, and




pressure profiles from radiosondes. We calculate the percentage of negative lifted index
above each station, which represents the occurrence rate of latent instability that exists
in the daytime (Figure 12c). The stations with high probability of strong instability,
denoted by P(lifted index $< 0$), are predominantly dispersed over the west U.S, the
west and south of China, and the Pacific Ocean, reaching a percentage as high as around
70%. These stations are regularly overlapped with great biases in the reanalysis
products as shown in Figures 5-8. According to the analysis, it is clear that all four
reanalysis products are positively associated with P (lifted index $< 0$) , with
correlation coefficients ranging from 0.34 to 0.47 (Figure 12d). The positive (negative)
correlation coefficients in lifted index suggests that the underestimation by reanalysis
might be associated with the instability activity in the lower troposphere that has not
been adequately represented or simulated by the models used in reanalyses. In light of
the surface heating during the day and the growth of the PBL due to air ascent, it is also
inferred that afternoon BLHs suffer the greatest errors if this is caused by inadequate
air mixing within the free troposphere in models.
**4. Conclusions and summary**
A climatology of near-global BLH from high-resolution radiosonde measurements
has been yielded for the daytime BLH. The high-resolution radiosonde data has a much
finer spatial resolution of 5 m or 10 m, compared to that by IGRA, and can establish a
finer and more precise structure of the PBL. In addition, direct comparisons among four
well-established reanalysis model products have been conducted. The present study
adopts over 300 sounding stations with high-resolution, spanning from 2012 to 2019,
to investigate the climatological variation of near-global BLH in the daytime and
evaluates four model products at the radiosonde sampling.
Notable spatial variation can be observed in the climatological mean of BLH at
0000 and 1200 UTC. In the afternoon, the regions over the Western United States and
Western China have the largest BLHs with values as high as 1.7 km, whereas 0000 and





1200 UTC compare generally to earlier times of day (LST) in the rest of the world with hence lower BLH. In addition, BLHs in the middle and low latitudes are larger than those in high latitudes. The $T_{2m}$ and RH, and the normalized sensible/latent heat flux are a good predictor for the spatio-temporal evolution of BLH. The most important result is we found that all the four reanalysis products generally underestimate the daytime BLH, with a near-global mean varying from around 130 m to 420 m. The largest bias in reanalysis appears over the Western United States and Western China, where the boundary layers grow vigorously in the afternoon. ERA-5 and MERRA-2 definitely have better performance than JRA-55 and NCEP-2 in terms of the magnitude of BLH and a higher correlation coefficient with the soundings. The newest version of reanalysis, ERA-5, has the smallest bias and the highest positive correlation relative to radiosondes. The underestimation by NCEP-2 and JRA-55 is robust over some regions, for instance, western China and western U.S, with differences even exceeding 800 m. However, all products can obtain a precise estimate over some regions, for instance, Europe, the eastern U.S, and polar, probably due to morning LST soundings and smaller daytime PBL development. The BLH over the Pacific Ocean is underestimated in all seasons and by all products. The underestimation tends to have a seasonal dependence, i.e., the warm season has a larger underestimation.

We investigated two possible sources contributing to the biases, including topography and static stability. The analysis shows that the DEM spread does have a positive correlation with the bias, suggesting that the reanalysis data cannot provide a reliable simulation result under complex terrain conditions. In addition, reanalysis BLH errors tends to be positively correlated with the occurrence rate of unstable air, suggesting that the reanalyses do not accurately determine BLH when the lower troposphere is unstable.

Although this study suffers from the inhomogeneous distribution of the radiosonde sites, the climatological BLHs at the near-global scale can help us understand the variation characteristics of BLH in different regions and for different LST. For the first time, we present near-global BLH estimates from high-resolution radiosondes, and



further conduct a comprehensive comparison of BLH products for four widely used
reanalysis datasets using the BLHs derived from the soundings. The findings provide
insights into the limitations of reanalysis data and, more importantly, are expected to
greatly benefit future research works related to applications of different kinds of
reanalysis data in the future.

**Acknowledgements**
This study is jointly supported by the National Key Research and Development
Program of the Ministry of Science and Technology of China under grant
2017YFC1501401, the National Natural Science Foundation of China under grant
41771399, 41531070 and 41874177, and S&T Development Fund of CAMS
(2021KJ008). The authors would like to acknowledge the National Meteorological
Information Centre (NMIC) of CMA, NOAA, German Deutscher Wetterdienst
(Climate Data Center), U.K Centre for Environmental Data Analysis (CEDA), GRUAN,
and the University of Wyoming (http://data.cma.cn/en,
ftp://ftp.ncdc.noaa.gov/pub/data/ua/data/1-sec/, https://cdc.dwd.de/portal/,
https://catalogue.ceda.ac.uk/,ftp://ftp.ncdc.noaa.gov/pub/data/gruan/processing/level2/
RS92-GDP/version-002/, http://weather.uwyo.edu) for providing the high-resolution
sounding data. We would like to thank the ECWMF for ERA-5 data
(https://cds.climate.copernicus.eu/cdsapp#!/dataset/reanalysis-era5-single-
levels?tab=form), GMAO for MERRA-2
(https://disc.gsfc.nasa.gov/datasets?keywords=MERRA-2&page=1), NCAR and Japan
Meteorological Agency for JRA-55 (https://climatedataguide.ucar.edu/climate-
data/jra-55), NOAA for NCEP-2
(https://psl.noaa.gov/data/gridded/data.ncep.reanalysis2.html). NASA for 30 arc
second digital evaluation height (DEM) data (https://search.earthdata.nasa.gov/).



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



**Figures:**

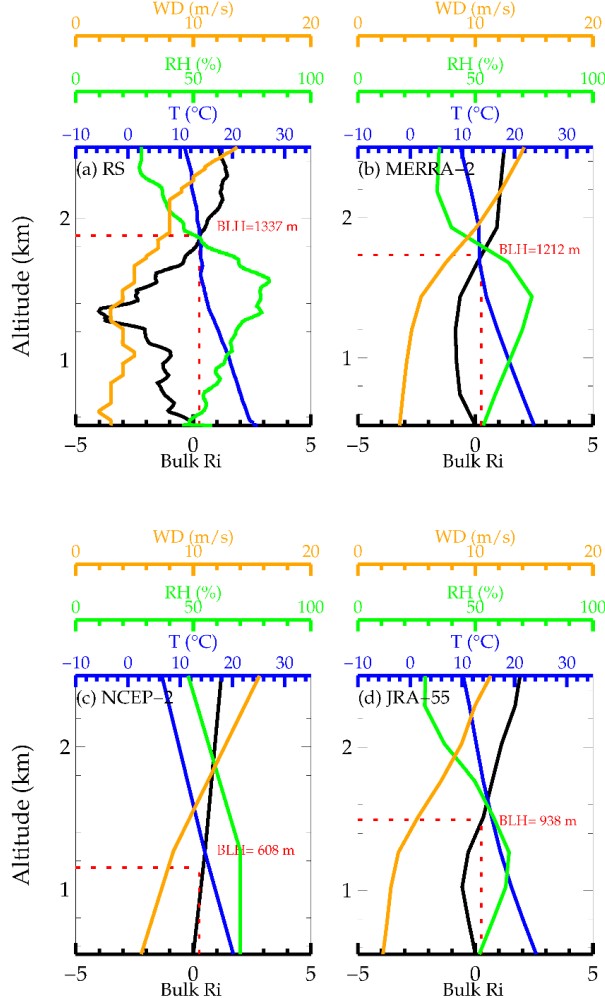

**Figure 1**. Profiles of basic atmomospheric parameters from the ground up to 2.5 km AGL, including wind speed (orange), bulk Ri (black), temperature (blue), and RH (green) at 0500 UTC (13 LST) 06 Jun 2016 at Chongqing (29.6°N, 106.4°E) from radiosonde (a), MERRA-2 (b), NCEP-2 (c), and JRA-55 (d) reanalysis datasets. Note that the boundary layer height in each subplot is marked by red dash lines and red texts, and the BLH for ERA-5 is 1265m in this case.



769

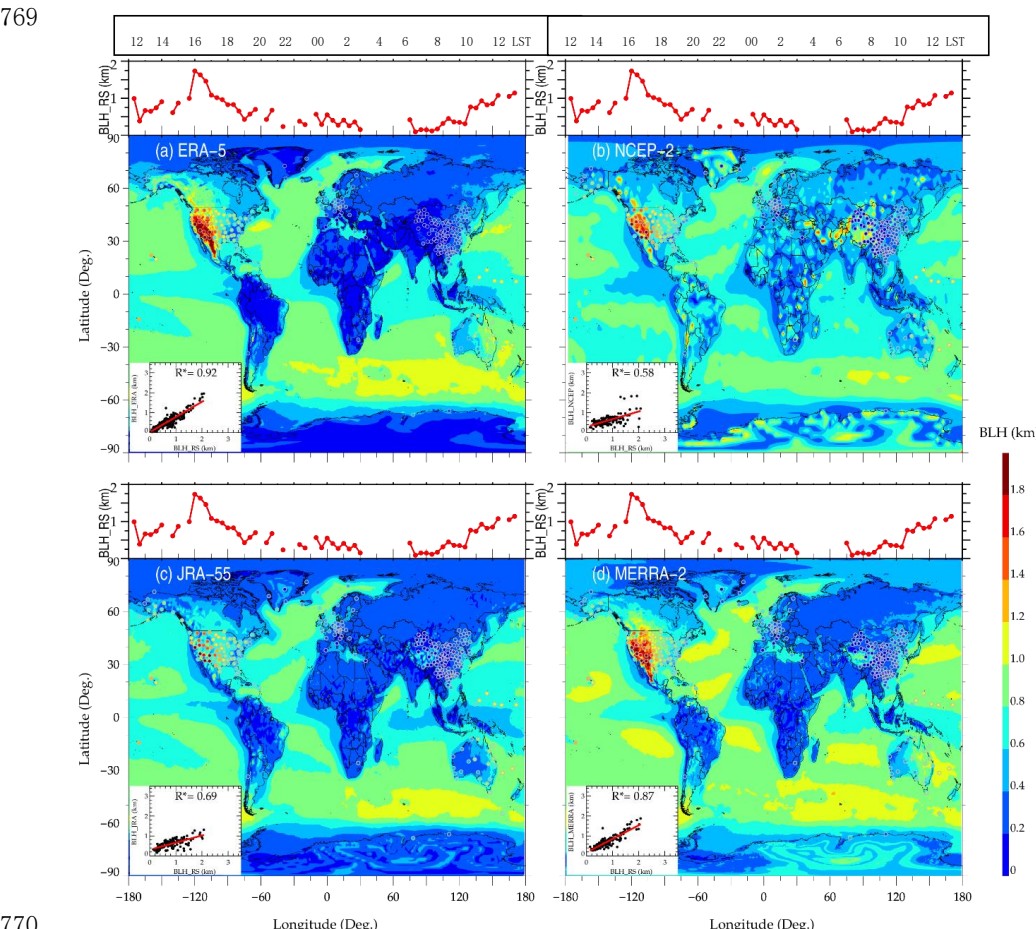

770

**Figure 2**. The ensemble mean BLH estimated from ERA-5 (a), NCEP-2 (b), JRA-55

(c), and MERRA-2 (d) reanalysis data at 0000 UTC during years 2012 − 2019. The dots

with gray marginal lines in each map denote the mean BLH derived by sondes at 0000

UTC, and the red dotted lines present the mean BLH derived by radiosonde on a grid

with 5° longitude. Stations with less than 10 profiles are not included in the analysis.

The 2D scatter plot in the left bottom corner of each panel illustrates the correlations

between reanalysis-derived and sonde-derived BLHs at 0000 UTC, where the asterisk

(*) superscripts indicate that the correlation coefficients are statistically significant

($p < 0.05$) and the red lines denote the least-squares regression line.

780



781

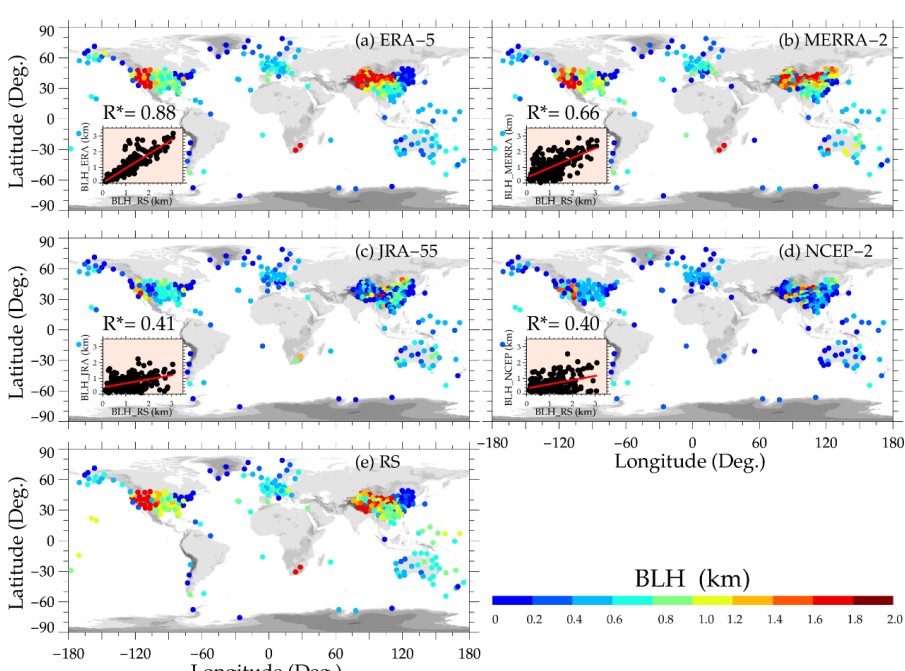

782

**Figure 3**. Spatial distributions of the mean BLHs determined at the near-global high-

resolution radiosonde observational network locations during the daytime for the period

2012 to 2019, which is extracted from ERA-5 (a), MERRA-2 (b), JRA-55 (c), NCEP-

2 (d), and radiosonde measurements (e), respectively. Similar to Figure 2, the scatter

plot illustrates the correlations between reanalysis-derived and sonde-determined BLHs

in the daytime.

789

790

791





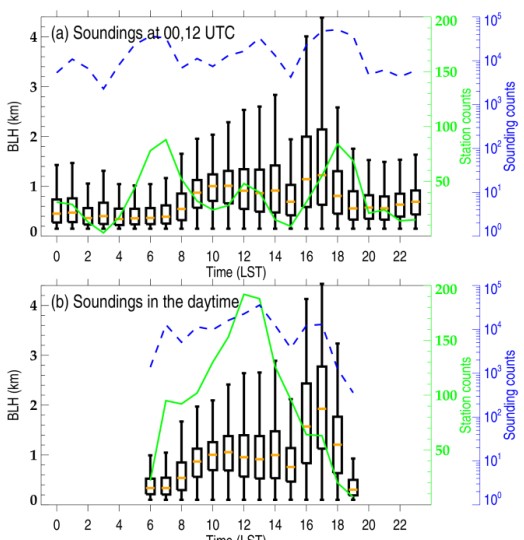

**Figure 4.** Box and whisker plots of diurnal variation (in LST, 24 hours) of BLH determined by all soundings operationally launched at 0000 and 1200 UTC (a) and by the soundings launched at both synoptic times and intensive observation times that are limited to the daytime alone (b). Solid green line and dotted blue line highlight the number of sonde station and total sounding for each hour of day, respectively.

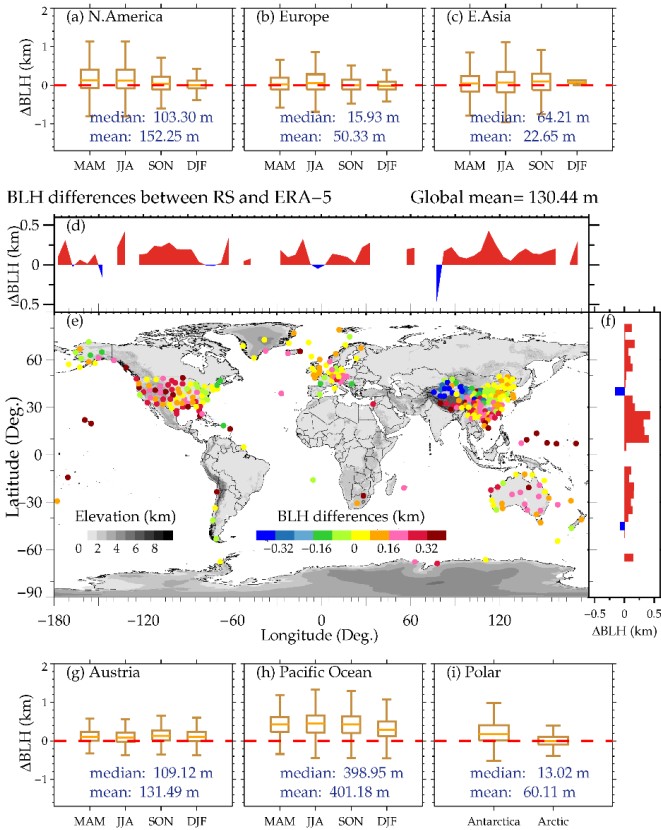

**Figure 5**. Statstical results of BLH differences between radiosonde and ERA-5. The spatial distribution of mean differences is highlighted in (e). Also shown are the distributions of mean BLH differences as a function of longitude (d) and latitude (f). The box and whisker plot of BLH differences over the six regions of interest (i.e., North America, Europe, East Asia, Austria, Pacific Ocean, Polar) over four seasons are displayed in (a-c), (g-i). The seasons are defined as follows: MAM, March–April–May; JJA, June–July–August; SON, September–October–November; DJF, December–January–February.

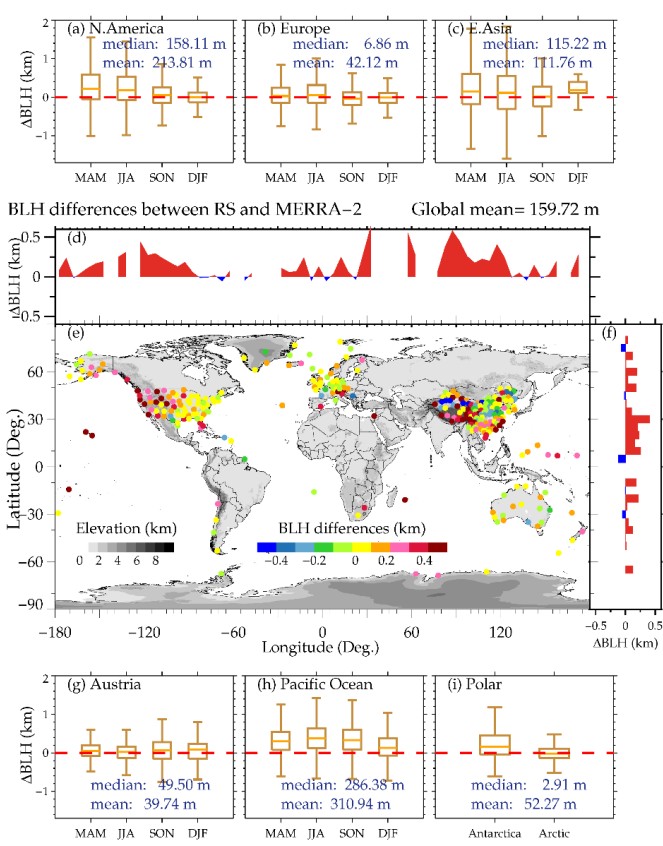

**Figure 6**. Similar as Figure 5, but for the differences between radiosonde-determined
BLHs and MERRA-2-derived BLHs.

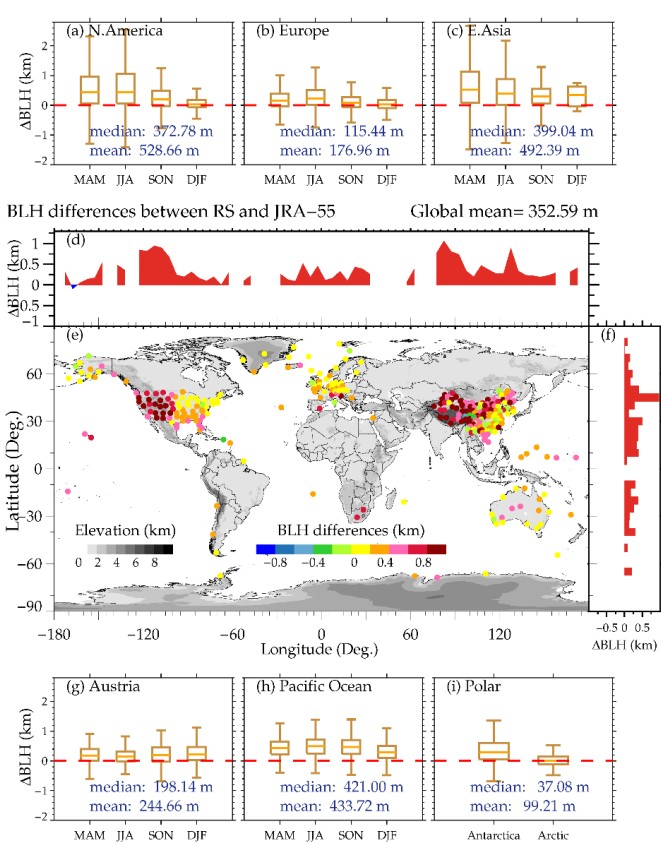


**Figure 7**. Similar as Figure 5, but for the differences between radiosonde-determined

BLHs and JRA-55-derived BLHs.













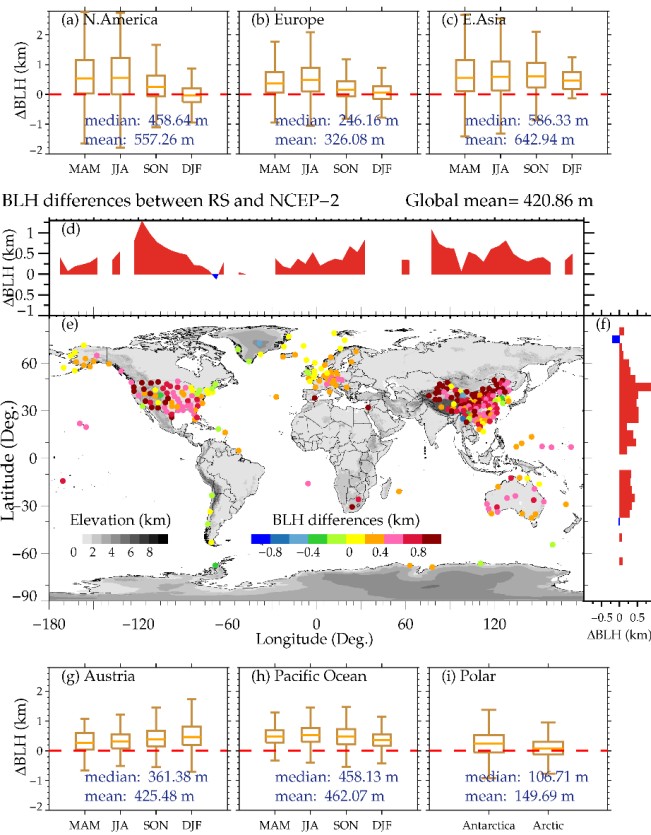


**Figure 8**. Similar as Figure 5, but for the differences between radiosonde-determined

BLHs and NCEP-2-derived BLHs












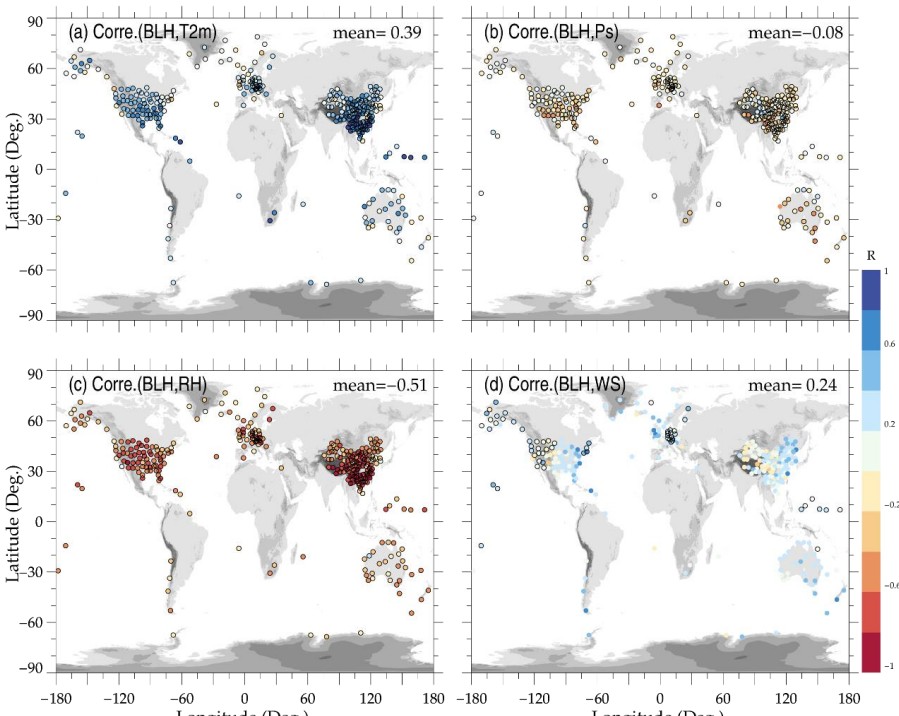


**Figure 9.** Correlations between the radiosonde-derived BLHs and near-surface air temperature at 2m AGL ($T_{2m}$; a), near-surface pressure (Ps; b), near-surface RH (c), and near-surface wind speed (WS; d). Dots outlined in black denote that the correlation coefficient values are statistically significant ($p<0.05$), and the mean correlations are texted in the upper right corner of each panel.














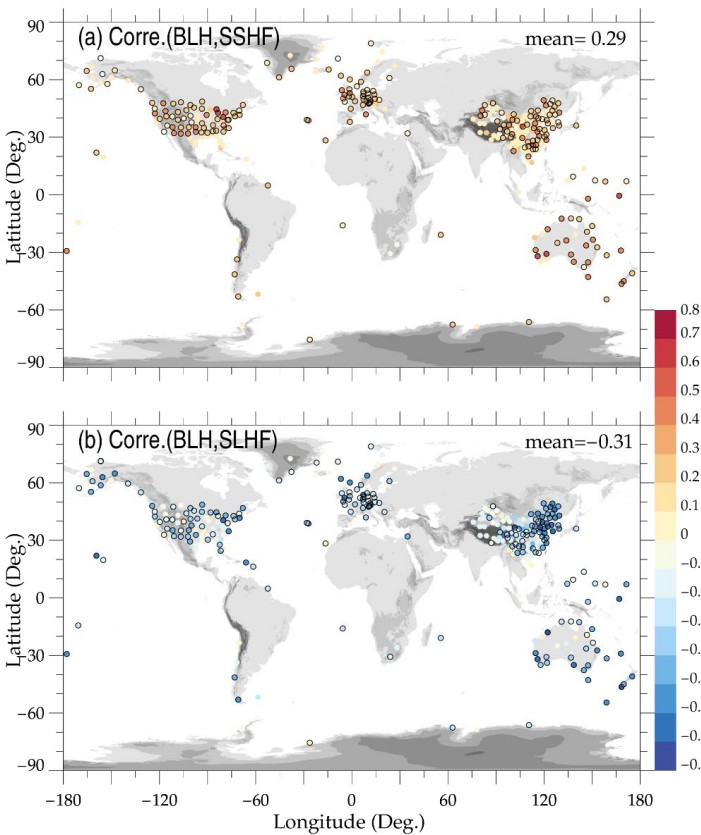


**Figure 10.** Similar as Figure 8, but for the correlations between BLHs versus normalized surface sensible (a) and latent heat fluxes (b).





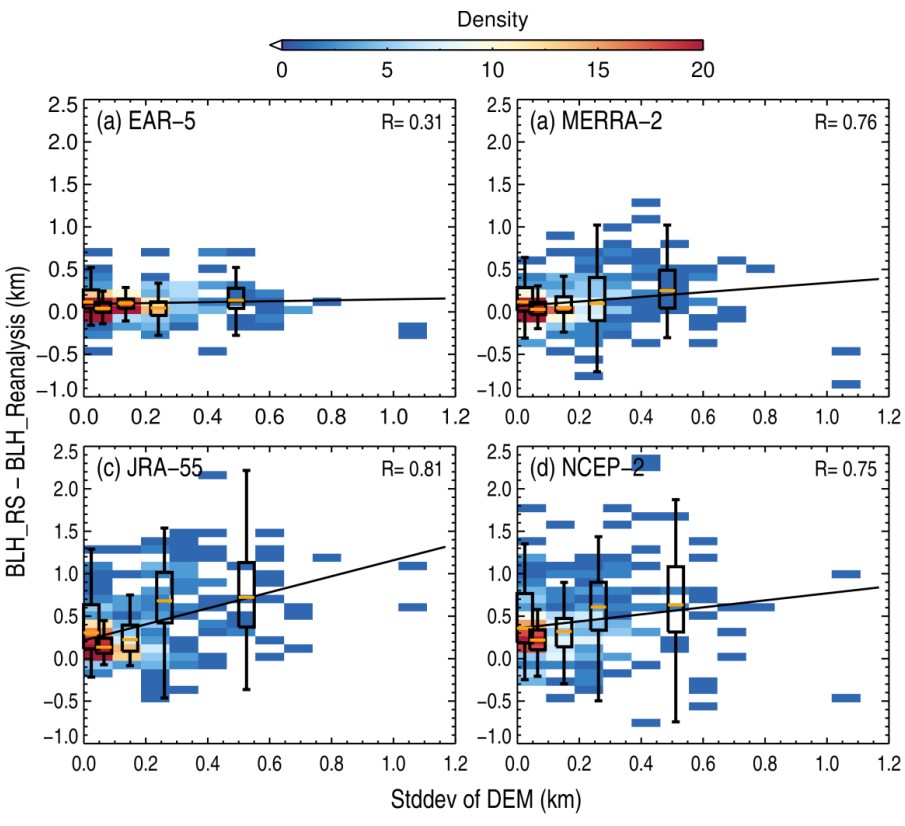

**Figure 11**. Density plots of the differences of BLHs between radiosonde and ERA-5
(a), MERRA-2 (b), JRA-55 (c), and NCEP-2 (d) as a function of the standard derivation
of the DEM, where the black lines denote the least-squares regression line. The box-
and-whisker plots of the anomalies of BLH in five evenly intervals are overlaid in each
panel, and the correlation coefficients are marked in the upper right corner of each panel.



894

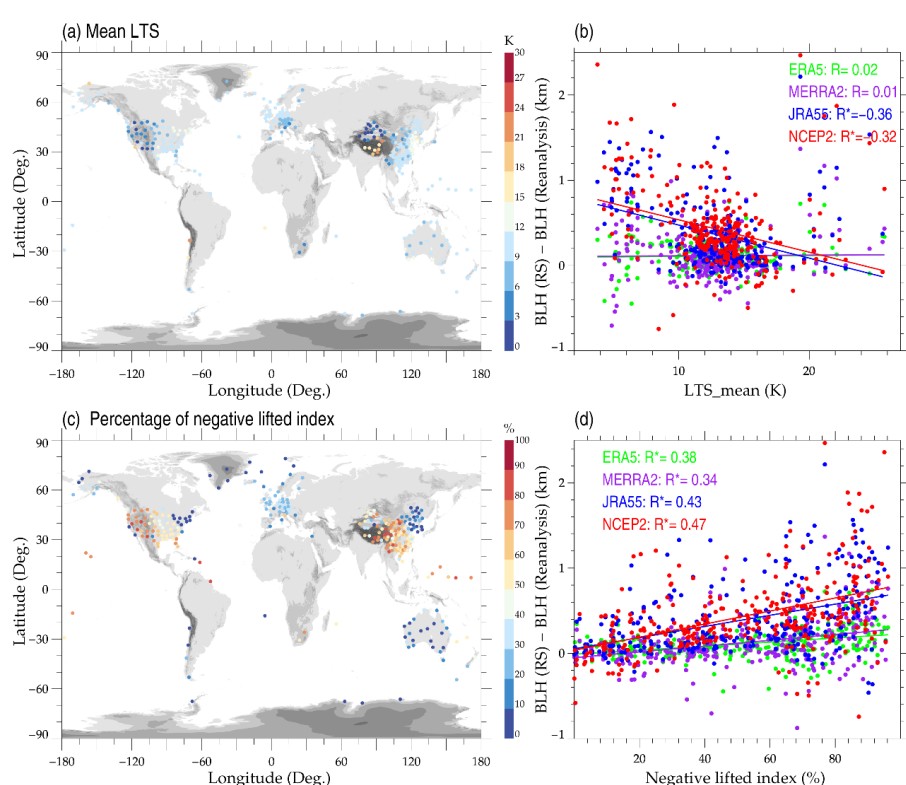

895

**Figure 12**. Spatial distribution of the ensemble means of lower tropospheric stability in the daytime (a). The scatter plots showing the difference of sounding- minus model-derived BLHs from four reanalysis datasets versus the anomalies of LTS as derived from four reanalysis relative to those from soundings (b). The variations in the percentage of negative lifted index (c), and the anomalies of BLH as a function of negative lifted index (d).