# Peer review of "Investigation of near-global daytime boundary layer height"

_Atmospheric Chemistry and Physics, 2021_

## Author Comment (AC1)

**Response to Reviewers' Comments**

*Before addressing the comments, we thank the editors and two (or three) anonymous reviewers for their thoughtful and constructive comments and suggestions, which significantly help improving the quality of our manuscript. In this revised manuscript, we have tried our best as much as possible to address all concerns and have revised the manuscript accordingly. The reviewers' comments are written in plain font, and our point-to-point responses to the reviewers' comments are in italics.*

**Reviewer #2 Evaluations:**

"Investigation of near-global daytime boundary layer height using high-resolution radiosondes: First results and comparison with ERA-5, MERRA-2, JRA-55, and NCEP-2 reanalyses" provides validations of simulated boundary layer heights from four commonly used reanalysis products on a near-global scale. The manuscript is nicely organized and comprehensive. Given the important role of reanalysis products in climatological analyses, in energy-focused resource assessments, and as inputs to higher-resolution models, validations such as the one presented here are essential for understanding reanalysis biases and limitations. Comments and suggestions for enhancement of this manuscript
follow.

***Response:*** *Many thanks for your positive recommendations. The concerns have been addressed as possible as we can in this revised manuscript.*

The discussion on the vertical resolution limitations of IGRA and the reanalysis products (Lines 96-99) would improve by including the numerical vertical resolutions (exact, on average, or a range) for each of these products, instead of simply stating that

they are sparse. This information is provided in Section 2, but since the manuscript defines the resolution of GPS RO on Line 92 it would be helpful for comparison to have this information for IGRA and the reanalyses in this location as well.

***Respons:*** *Per your kind suggestion, the statement has been revised as:*

*"Compared with high-resolution soundings, IGRA is sparsely sampled in the vertical (about 10-30 layers below 500 hPa), which could result in large uncertainties in estimating BLH. Likewise, additional errors could be introduced in reanalysis products for their sparse vertical resolutions (about 6-42 layers below 500 hPa), which are equivalent to or bigger than IGRA."*

The authors are disregarding the packaged BLH parameter from MERRA-2 and recalculating BLH in a more similar fashion to the ERA5 definition (Lines 202-207). For the benefit of reanalysis users, it is highly recommended that the MERRA-2 packaged BLH parameter is also validated along with the author-derived version. Can this comparative analysis be included?

***Response:*** *Good point! Following your thoughtful comments, the related contents have been added to this revised manuscript. As a matter of fact, the packaged BLH (in unit Pa) in MERRA2 is defined by the critical value of* heat diffusiv*ity which is different from the method used in present analysis. The results in Figs. S3,S4 show that the packaged BLH in MERRA2 is considerably overestimated by around 0.8 km over eastern China. BLH over other regions are slightly or moderately overestimated by around 50 m.*

"In this product, the BLH is packaged and defined by identifying the lowest level at which the heat diffusivity drops below a threshold value (McGrath-Spangler and Denning, 2012). The formula for calculating BLH is as follows:

$$\text{BLH(MERRA2\_packaged)} = 44308 \times (1 - \left(P_{PBLtop}/P_{Surface}\right)^{0.1903} \quad (1)$$

where $\text{BLH(MERRA2\_packaged)}$ is in unit of meter, $P_{PBLtop}$ the BLH (packaged

parameter in MERRA-2, in unit Pa), and $P_{Surface}$ the surface pressure (in unit Pa). However, to preclude the uncertainty raised by different methods adopted, the BLH by MERRA-2 is extracted by bulk Richardson number method, by utilizing the parameters of horizontal wind, temperature, geopotential height, relative humidity (RH), and surface pressure as inputs. These input data are provided on a grid of 576×361 points with 0.625° longitude and 0.5° latitude resolution and has 42 pressure levels (about 16 layers below 500 hPa), with a temporal resolution of 3 h.*"*

*"In addition, the packaged BLH in MERRA-2 is also evaluated with radiosonde. BLH is as high as 3 km over the TP region at 0600 UTC (Figure S3), corresponding to an overestimation of 0.8 km over this region (Figure S4). Over the rest regions, BLH is slightly or moderately overestimated by around 50 m. However, The BLH difference among various methods could reach up to a kilometer or even more (Seidel et al., 2010), which is probably owing to the variety of kinetic or thermodynamic theories applied in different algorithms."*

[Figure]

***Figure S3***. *The mean packaged BLH in MERRA-2 at (a) 0000 UTC, (b) 0600 UTC, (c) 1200 UTC, (d) 1800 UTC. The dots with gray marginal lines in each map denote the mean BLH derived by sondes.*

[Figure]

***Figure S4***. *Differences between BLH(RS) and* `BLH(MERRA2_packaged)`. *The spatial distribution of mean differences is highlighted in (e). Also shown are the distributions of mean BLH differences as a function of longitude (d) and latitude (f). The box and whisker plot of BLH differences over the six regions of interest (i.e., North America, Europe, East Asia, Australia, Pacific Ocean, Polar) over four seasons are displayed in (a-c), (g-i). The seasons are defined as follows: MAM, March–April–May; JJA, June–July–August; SON, September–October–November; DJF, December– January–February.*

Figure 1 is a helpful case study to assist the reader in the methodology. Could ERA5 be included in the graphic as well, instead of a brief mention in the figure caption?

***Response:*** *Per your suggestion, in the present analysis, we use the packaged BLH in ERA5 since it is estimated by the bulk Ri method. In the revised Fig.1, we added black dash lines to mark the BLH derived from ERA5.*

Section 3.3 provides an interesting attempt to correlate BLH with near surface measurements, however it should be moved to a different location in the manuscript, as it does not involve the reanalysis products and therefore does not flow with the surrounding sections. Additionally, the enthusiasm over the correlation results in this section should be tempered. 0.39 is not a "relatively high positive correlation coefficient". Perhaps "moderate" might be a better choice.

*Response: Per your suggestion, section 3.3 has moved forward as section 3.2. The phrase has been modified as:*

*"Moderate positive (negative) correlation coefficients can be noticed between BLH and $T_{2m}$ (RH), with mean values of 0.39/-0.51 (Figure 5a, c)…".*

I second Anonymous Reviewer #3 in suggesting that presenting results according to reanalysis minus radiosondes is much more easily understood that radiosondes minus reanalysis.

*Response: Amended as suggested.*

Specific comments:

Line 33: Suggest adding "analysis" after "air quality, weather and climate".

*Response: Amended as suggested.*

Line 85: Suggest rewording "And notable diurnal and seasonal cycles have been revealed" to Notable diurnal and seasonal cycles in BLH variation have been revealed".

*Response: Point taken.*

Line 113: Elaborate numerically on "a rough consistency".

*Response: As suggested, the sentence has been rephrased as:*

*"Some inter-comparisons between instruments or model data, such as radiosonde, CALIOP, and ERA-interim reanalysis have been previously conducted, and a good*

*consistency has been yielded in seasonal and spatial variation (e.g., Guo et al., 2016; Zhang et al., 2016)."*

Line 184: Suggest rewording "As a result, ..." to "Using this definition, 190,013 profiles including soundings launched at both synoptic times and during IOP, spanning January 2012 to December 2019, are used to obtain the BLH in the daytime."

***Response:*** *Done as suggested.*

Line 190: Reword "undergo" to "has undergone".

***Response:*** *Point taken.*

Lines 205, 382, 384, 460: Change "MERR-2" to "MERRA-2".

***Response:*** *Amended as suggested.*

Line 213: Give NCEP-2 its own paragraph beginning here, as was done for the other reanalysis products.

***Response:*** *Point taken.*

Line 363 and Figures 5-8: "Austria" should be "Australia"?

***Response:*** *Yes. Very thanks for the correction. All fixes have been made.*

Line 377: Recommend numerically describing the seasonal differences in the bias.

***Response:*** *As suggested, it has been rephrased as:*

*"The bias seems to exhibit a seasonal dependence, and it is around 62 m larger in the warm seasons compared to cool seasons in both hemispheres."*

Line 394: Reword "acceptable" to "more in line with the observations". Acceptable is too subjective.

***Response:*** *Point taken.*

Line 425, Figure 11: Change "EAR-5" to ERA5.

**Response:** *Point taken.*

---

## Author Comment (AC3)

**Response to Reviewers' Comments**

*Before addressing the comments, we thank the editors and two (or three) anonymous reviewers for their thoughtful and constructive comments and suggestions, which significantly help improving the quality of our manuscript. In this revised manuscript, we have tried our best as much as possible to address all concerns and have revised the manuscript accordingly. The reviewers' comments are written in plain font, and our point-to-point responses to the reviewers' comments are in italics.*

**Reviewer #3 Evaluations:**

The manuscript entitled "Investigation of near-global daytime boundary layer height using high-resolution radiosondes: First results and comparison with ERA-5, MERRA-2, JRA-55, and NCEP-2 reanalyses" presents a near-global assessment of high-resolution radiosonde derived boundary layer height (BLH) and provides a quantitative assessment of four reanalysis products. This paper is generally well written and makes an important contribution to characterizing the BLH at the global scale and providing useful information on reanalysis data usage. However, I have the following major comments concerning the bias attribution.

*Response: We greatly appreciate your positive comments on the contribution of our work to the PBL meteorology, especially in characterizing the BLH across the world, as well as comprehensive evaluation of several widely used reanalysis dataset based on high-resolution radiosonde measurements. Per your suggestion, we have tried our best to address all your concerns in this revised manuscript, which we hope you will be satisfied with.*

First, in the case study at Chongqing, the fine-scale vertical structures of Ri, WS, RH, and T seem to have a larger impact in determining BLH compared to the overall bias

of the basic parameters. It appears that both overestimation (in JRA-55) and underestimation (in NCEP-2) of WS and RH lead to a smaller BLH. Discussions on the impact of vertical structure including the vertical resolution would provide useful information on the bias attribution. Relatedly, is there a specific reason for choosing this case as an example to show BLH biases in the reanalysis data? It would be helpful to provide a comment on other cases.

***Response:*** *Strongly agreed. Based on Eq.(2), BLH is negatively correlated with wind speed (WS), relative humidity (RH) and temperature profiles. Particularly, it is largely altered by the near-surface meteorological parameters and the vertical resolution of data. Based on the result in Seidel et al. (2012), BLH is usually lower for a sparser vertical resolution. Factors that cause uncertainties in estimating BLH by using Richardson method include, but not limited to, meteorological parameters, the surface friction, vertical resolution of data and the critical value of Ri.*

*Compared to vertical profiles of RH, temperature and wind speed, BLH considerably varies with the near-surface virtual potential temperature. In Figure 1, the near-surface virtual potential temperatures are underestimated by MERRA-2, JRA-55 and NCEP-2 ($\theta_{vs}(RS) = 304.43\ K, \theta_{vs}(MERRA2) = 303.21\ K, \theta_{vs}(NCEP2) = 301.97\ K, \theta_{vs}(JRA55) = 303.59\ K$).*

*The reasons for the selection of site located in Chongqing were twofold: (a) The elevation of Chongqing station is 541 m above sea level, which is a typical value of elevation among all radiosonde stations. It is therefore helpful for examining the impact of surface parameter extraction procedure that is an important input parameters for the BLH from Ri method. (b) The time for the balloon launch is at 0600 UTC (1300 LST) when convective PBL dominates. This justifies our selection of this case, to some extent.*

*In addition, we investigate three more cases, as illustrated in Figs. A-C. It is found that the underestimations in BLH could be mostly owing to the smaller values of $\theta_{vs}$, wind speed, and RH, as well as the coarser vertical resolution. The aforementioned response and comments have been well incorporated into this revised manuscript.*

[Figure]

*Figure A*. *Profiles of basic atmomospheric parameters from the ground up to 2.5 km AGL, including wind speed (orange), bulk Ri (black), temperature (blue), and RH (green) at 0600 UTC (1400 LST) 18 Aug 2015 at Beijing (39.8°N, 116.46°E) from radiosonde (a), MERRA-2 (b), NCEP-2 (c), and JRA-55 (d) reanalysis datasets. Note that BLH derived from ERA5 is denoted by black dash lines.*

[Figure]

**Figure B**. *Similar to Fig. A, but for sounding at 1800 UTC (1100 LST) 10 Jul 2019 at CORPUS CHRISTI (27.77°N, -97.5°W).*

[Figure]

***Figure C****. Similar to Fig. A, but for sounding at 0600 UTC (1300 LST) 10 Aug 2018 at KOROR/PALAU ISLAND (7.33°N, 134.48°E).*

Second, the biases of the BLH in reanalysis data are attributed to the complex topography and static stability based on their correlation coefficient. The afternoon sounding during the warm season leads to large biases over the TP and western US, where the terrain is complex. Assessing the relationship between BLH bias and DEM spread using data collected at similar LST would provide useful information on the robustness of the results.

***Response:*** *We agree. Following your constructive suggestion, we assessed the relationship between BLH bias and DEM spread only for all soundings released in the afternoon, spanning from 1300 LST to 1800 LST. As a result, there were 78 available radiosonde stations in total. As presented in Fig. D (Fig. 11 in the revised manuscript),*

*BLH biases are still negatively correlated with DEM spread, indicating a robust relationship between them. The related statement has been rephrased as:*

*"Terrain is complex over the western China and western US where most of soundings are released at afternoon and large BLH biases are usually found. Therefore, for all soundings that are launched at the time interval spanning from 1300 LST to 1800 LST we analyze the relationship between BLH biases and the standard derivation of the DEM (Figure 11)."*

[Figure]

***Figure 11**. Density plots of the BLH biases in ERA-5 (a), MERRA-2 (b), JRA-55 (c), and NCEP-2 (d) as a function of the standard derivation of the DEM. All samples are collected from soundings that are launched in the afternoon, spanning from 1300 LST to 1800 LST.*

Meanwhile, because of the coarser temporal resolution, MERRA-2, JRA-55, and NCEP-2 are not able to match LST of all soundings during IOP. The time mismatch between the sounding and reanalysis data may also introduce biases due to the distinct diurnal variation of BLH. It is necessary to discuss if the result will significantly change with/without IOP data used.

***Response:** Good points! Per your suggestion, we plotted the distribution of BLH by*

*using soundings that were released only at 0000 UTC and 1200 UTC (Fig. D). Compared to the result in Fig.3, the result will not significantly change with/without IOP data used. It can be interpreted by the fact that about 75% of soundings were released at regular synoptic time.*

[Figure]

***Figure D.** Spatial distributions of the mean BLHs determined at the near-global high-resolution radiosonde observational network locations during the daytime (without IOP obervations) for the period 2012 to 2019, which is extracted from ERA-5 (a), MERRA-2 (b), JRA-55 (c), NCEP-2 (d), and radiosonde measurements (e), respectively.*

Fig. 4 nicely shows the diurnal variation of BLH. The authors mention "some soundings that are released at 0000 and 1200 UTC are excluded …. for collecting samples in the daytime." In my understanding, for instance, the 14 LST results in both Fig.4a and Fig. 4b should include all soundings collected at 14 LST. It is not very clear why some soundings at 0000 and 1200 UTC are removed to only show daytime results in Fig. 4b? Besides, how does the application of additional soundings during IOP lead to the differences between Figs. 4a and 4b?

***Response:*** *As shown in Fig.2, soundings released over China and Europe at 0000 UTC are during nighttime. In addition, Fig.S2 shows the result for 1200 UTC. As a result, only 21.38% percent of sounding at 0000 and 1200 UTC released in the daytime.*

*According to Fig. E, the durinal variation of BLH is insignificantly influenced by IOP obervations.*

*The aforementioned response and comments have been well incorporated into the revised manuscript.*

[Figure]

***Figure E.*** *(a) and (b) show the durinal variation of BLH in the daytime with and without IOP observations, respectively.*

Is there a specific reason for presenting the difference using radiosonde (the reference dataset) minus reanalysis rather than reanalysis minus radiosonde in Figs 5-8? It seems counterintuitive to use positive differences in those figures to represent underestimated BLHs.

***Response:*** *Good point! As suggested by both reviewers, Figs.7-10 in the revised*

*manuscript have been modified as "BLH(reanalysis) – BLH(RS)".*

Specific Comments:

Line 56: Suggest changing to "boundary layer height".

*Response: Amended as suggested.*

Line 192: How many layers below 500 hPa in ERA-5?

*Response: ERA-5 in versions of pressure level and model level has 16 and 42 layers below 500 hPa, respectively. While the used BLH in this manuscript is the packaged product, which can be found at https://cds.climate.copernicus.eu/cdsapp#!/dataset/reanalysis-era5-single-levels?tab=form*

Line 218: Change to 0000 and 1200 UTC.

*Response: Point taken.*

Line 220: This section introduces calculations for both normalized sensible heat and latent heat fluxes. Suggest changing the section title to include both fluxes.

*Response: Point taken.*

Line 225: Add a period after the parenthesis. Can you further explain why small latent heat flux means more energy being available for PBL growth?

*Response: Point taken. As suggested, a new phrase has been added as "When less energy is constrained by the moist ground, more energy is available to heat the air.".*

Line 236: Remove "sensible".

*Response: Point taken.*

Line 237: Sections 2.4 and 2.5 introduce BLH calculation which may be more

connected to section 2.2. Suggest moving those two sections forward.

*Response: Point taken.*

Line 272-273: Is this an extra step only required by observations during IOP, as the regular synoptic times are included in all reanalysis data? Meanwhile, JRA-55 and NCEP-2 have a temporal resolution of 6 hours, which may be not able to hit every weather balloon launch time with hour difference. Would it result in a significantly smaller sample size compared to ERA-5 and MERRA2?

*Response: Yes, step (3) is only suitable for IOP observations. As a result, the samples by JRA-55 and NCEP-5 are indeed less than those of ERA-5 and MERRA2. The total number of samples for NCEP2 and JRA55 are 18.37% less than that of ERA5, which would not significantly change the result.*

Line 282: Is there a specific reason for arranging the panels in the order of a, b, d, c?

*Response: There is no specific reason. Per your suggestion, we have changed it to ordinary order.*

Line 345-346: The authors mentioned that the reanalyses and observations show the deepest BLH in the afternoon of summer, from which I think it is insufficient to conclude that "both capture the diurnal and seasonal variations" at this point.

*Response: We totally agree and revise it to "By and large, the climatological results of BLH by radiosonde and four model products are comparable, indicating that both capture the spatial variations implied by the sounding LST times sampled."*

Line 365-366: Did the authors mean "latitude" and "67.6 °N/°S"?

*Response: Yes. The error has been corrected.*

Line 385: Remove "/.".

*Response: Point taken.*

Line 392: What is the "ensemble mean"?

*Response: We changed it to near-global mean.*

Line 413: Change "WD" to "WS", and at other places.

*Response: Point taken.*

Line 423: Fig. 9b marks significant correlations between BLH and Ps. I think this was simply left out by mistake.

*Response: Yes, we agree. Per your comment, it has been corrected as: "By contrast, the correlation between Ps and BLH is negatively significant above most of the regions (Figure 5b)"*